# Measurement of Physical Activity by Shoe-Based Accelerometers—Calibration and Free-Living Validation

**DOI:** 10.3390/s21072333

**Published:** 2021-03-26

**Authors:** Jonatan Fridolfsson, Daniel Arvidsson, Stefan Grau

**Affiliations:** Center for Health and Performance, Department of Food and Nutrition, and Sport Science, Faculty of Education, University of Gothenburg, SE-405 30 Gothenburg, Sweden; daniel.arvidsson@gu.se (D.A.); stefan.grau@gu.se (S.G.)

**Keywords:** occupational health, cut-points, energy expenditure, biomechanics, workload, indirect calorimetry

## Abstract

There is conflicting evidence regarding the health implications of high occupational physical activity (PA). Shoe-based accelerometers could provide a feasible solution for PA measurement in workplace settings. This study aimed to develop calibration models for estimation of energy expenditure (EE) from shoe-based accelerometers, validate the performance in a workplace setting and compare it to the most commonly used accelerometer positions. Models for EE estimation were calibrated in a laboratory setting for the shoe, hip, thigh and wrist worn accelerometers. These models were validated in a free-living workplace setting. Furthermore, additional models were developed from free-living data. All sensor positions performed well in the laboratory setting. When the calibration models derived from laboratory data were validated in free living, the shoe, hip and thigh sensors displayed higher correlation, but lower agreement, with measured EE compared to the wrist sensor. Using free-living data for calibration improved the agreement of the shoe, hip and thigh sensors. This study suggests that the performance of a shoe-based accelerometer is similar to the most commonly used sensor positions with regard to PA measurement. Furthermore, it highlights limitations in using the relationship between accelerometer output and EE from a laboratory setting to estimate EE in a free-living setting.

## 1. Introduction

Measurement of physical activity (PA) by accelerometers is common practice in clinical and epidemiological research [1]. PA is generally associated with decreased risk of cardiovascular disease and reduced mortality [2]. However, in workplace settings PA and physical workload is associated with higher risk of sick leave as well as mental ill-health and all-cause mortality [3,4]. Since most of the research on occupational PA relies on self-reported measures of PA, more high-quality measurement of PA in workplace settings is required to understand this relationship further [3].

In PA measurement, PA is typically quantified as either time spent at different activity intensities (e.g., light or moderate) or activity types (e.g., sitting or walking) [1]. Measures of PA intensity are the most common and the reference used for intensity is energy expenditure (EE). PA is defined as “any bodily movement produced by skeletal muscles that results in energy expenditure” [5] and is more closely resembled by intensity measures. In these measures, bodily movement recorded as acceleration is aggregated or averaged over time to represent volume and intensity of activity [6]. This is followed by applying calibration algorithms to translate measured movement to EE [6]. Accelerometers are most often positioned on the hip or on the wrist [1,6]. In addition, thigh positioning has also been used for PA measurement [7,8]. However, when the validity of the most commonly used positions are compared in free-living conditions, the hip position consistently outperforms the wrist position whereas the wrist position seems to be superior to thigh [1,9].

Shoe positioning of accelerometers has recently been introduced in PA measurement [10,11]. Since safety-shoes are often required at workplaces, shoe positioning of accelerometers provides a suitable solution for PA surveillance. However, so far only activity type classification has been implemented for shoe-positioned accelerometers [10,11]. Nevertheless, this position could also be suitable for measuring PA intensity. The performance of shoe accelerometers with regards to quantifying volume and intensity of PA have not been investigated previously and no calibrations to EE are available. Further, validation of calibration algorithms in real workplace setting is rarely performed. Development of algorithms for PA intensity measurement from shoe-based accelerometers could facilitate investigation of the association between occupational PA and health.

This study aimed to develop calibration models for estimation of EE from shoe-based accelerometers and validate the performance in a workplace setting. In addition, the performance was compared to the most commonly used accelerometer positions.

## 2. Materials and Methods

### 2.1. Design

The study was conducted in two parts. First, a calibration part where participants performed structured activities wearing multiple accelerometers while EE was measured. Second, a free-living validation part where participants worked in their usual occupation wearing multiple accelerometers while EE was measured. Calibration models predicting EE from accelerometer output were developed based on data in the first part. In the second part, these models were validated using free-living data from industrial workers. Furthermore, additional calibration models were derived from the free-living data.

### 2.2. Participants

In the calibration part, 34 participants were recruited among staff and students from the Department of Food and Nutrition and Sport Science at the University of Gothenburg. In the validation part, participants were recruited among workers at an industrial factory. Fifteen participants were recruited among workers in industrial production and 14 participants were recruited among workers in logistics warehouse. Inclusion criteria were to be between 18 and 65 years of age, and having no physical limitation, medical condition or illness that would interfere with performing the activities in the calibration protocol and working as usual at the workplace, and affect the measurement of PA or energy expenditure. Written informed consent was retrieved from all participants and ethical approval was granted from the regional ethics committee in Gothenburg (No. 765–18).

### 2.3. Protocol

Participants in the calibration part performed five different activities each according to a structured protocol. Each activity was performed for four minutes in order to reach a steady state for oxygen consumption [12]. The activities were sitting, standing, slow walking, brisk walking and running. During sitting and standing, participants were instructed to solve Sudoku to closer simulate stationary work. The locomotive activities were performed outdoors at a self-selected pace.

Participants in the validation part were instructed to work as normal while their PA was measured. These measurements were performed for approximately 60 min. 

### 2.4. Energy Expenditure

EE was measured using a portable indirect calorimetry system worn on the back (Cosmed K5, Cosmed, Rome, Italy). Participants wore a mask covering the nose and mouth to capture exhaled air while oxygen consumption was measured breath by breath. EE was calculated as metabolic equivalents, which is the oxygen consumption of activity relative to the resting metabolic rate. Resting metabolic rate was estimated as 3.5 mL∙min^−1^∙kg^−1^.

### 2.5. Physical Activity

PA was measured by triaxial accelerometers (AX3, Axivity Ltd., Newcastle upon Tyne, UK), set to record acceleration with a sample rate of 100 Hz and a range of ±16 g. A range of ±6 g is typically used in PA measurement at the hip and wrist. However, 16 g is required for capturing PA at the shoe because of higher acceleration peaks during normal movement [13].

The accelerometer positions evaluated in the calibration and validation parts were hip, wrist, thigh and shoe. The hip and wrist accelerometers were attached to an elastic band positioned laterally above the right hip and left dorsal wrist respectively. The thigh accelerometer was attached to the mid right anterior thigh using medical grade adhesive film. These positions are normally used in clinical and epidemiological research [1]. All participants wore the same shoe model (Ergo-Active Grant, Elten GmbH, Uedem, Germany) that includes a rigid heel-cap. The shoe accelerometer was attached to the outside of the rigid heel-cap on the dorsal side of the right shoe with non-elastic tape.

Raw accelerometer data was processed to remove the influence of gravity and noise with the 10 Hz frequency extended method (FEM). FEM involves a band-pass frequency filter with high-pass and low-pass components at 0.69 and 10 Hz respectively [14]. This method has been shown to outperform the most commonly used method of data processing in PA research from a biomechanical and physiological perspective since it accurately captures the entire intensity range [14,15]. Output from the three axes were combined to a vector magnitude.

### 2.6. Mechanical Workload

As a reference for mechanical workload, nine additional accelerometers were attached to the body. These accelerometers were positioned to capture the movement of separate body segments considered rigid: shank, thigh, lower arm, upper arm and trunk including head. Accelerometers were positioned on both the left and right arm and leg laterally as close as possible to their respective center of mass [16]. The trunk accelerometer was positioned at the lower back. Accelerometer output from all sensors was processed as above. Processed output was weighted by the approximate segment weight relative total body weight [16]. Subsequently, weighted output from all accelerometers were summed up to represent total body mechanical workload.

### 2.7. Statistical Analyses

Calibrations between accelerometer output and EE were performed by applying smoothing splines to allow for non-linear relationships [7,14]. Smoothing splines were implemented with a normalized smoothing parameter of 0.2. Model fit was evaluated by explained variation (R^2^) in EE. In addition, the association between accelerometer output from the four positions investigated and total body mechanical work was evaluated by smoothing splines as above.

In the validation part, the calibration models derived from smoothing splines were applied to the accelerometer data collected from the investigated positions to estimate EE. Estimated and measured EE was compared by both participant mean and minute-by-minute. The mean estimated and predicted EE from the entire measurement period was compared for each participant. In order to compare EE minute-by-minute, measured EE was averaged minute-by-minute whereas estimated EE was averaged over two minutes with 50% overlap. Hence, one-minute mean measured EE was compared to a two-minute mean estimated EE from the same and previous minute. The rationale for this was that the measured VO_2_ response is not immediate in relation to mechanical work captured by the accelerometer. However, considering the same and previous minute acceleration (estimated EE) should be sufficient time for the measured EE to respond to the corresponding workload [12]. The association between predicted and measured EE was assessed by Pearson correlation (r) and visualized by scatter plots. Furthermore, the agreement was assessed by root mean squared error (RMSE) and visualized by Bland—Altman plots.

As an alternative solution to structured calibration, additional calibration models were derived from the free-living data. These calibrations were generated, as above, based on minute-by-minute accelerometer data and EE. The performance of the free-living calibrations was assessed by leave one out cross-validation (LOO). Data from all participants except one was used for calibration and the performance validated on the data from the participant left out. This was repeated until all participants had been left out for validation once. Free-living calibration was done both for data from participant mean and minute-by-minute.

Cut-points representing 1.5, 3, 6 and 9 metabolic equivalents (METs) were derived from the calibration models based on either structured or free-living data [2]. Differences in participants’ characteristics between the participant groups were investigated by independent t-tests and chi-square test for independence. All data processing and statistical analyses were performed in MATLAB R2020b (MathWorks, Natick, MA, USA). The data collected from laboratory and validation is available in Appendix A. 

## 3. Results

### 3.1. Participants Characteristics

Participants’ characteristics are presented in Table 1. There were significant differences (*p* < 0.05) between the laboratory participant group and the two workplace groups regarding age and BMI. Furthermore, there was a significant difference in the proportion of females between the laboratory group and the industrial production group.

### 3.2. Calibration

The explained variation (R^2^) of the smoothing spline-based calibration models between accelerometer output from the four sensor positions investigated and EE is presented in Table 2. Calibrations were developed separately on laboratory and free-living data. The calibration modes are visualized in Figure 1. R^2^ of all investigated positions was similar in the laboratory setting whereas in free living, the hip position was best and wrist position worst. The most commonly used accelerometer cut-points for EE, retrieved from the calibration models, are presented in Table 3. No cut-points above three METs were derived from the free-living data, since no data was available for calibration at this intensity. The cut-points derived from laboratory data were higher than the cut-points from the free-living data.

### 3.3. Validation

The association between EE estimated based on the laboratory calibration model and measured EE are shown in Figure 2. In addition, the agreement between estimated and measured EE are shown in Figure 3. Correlation and agreement are shown for both minute-by-minute measures and participant mean and are quantified in Table 2. The wrist position displayed the weakest correlation between estimated and measured EE. On the contrary, the wrist position displayed the highest agreement between estimated and measured EE. The correlation and agreement from the free-living calibrations based on LOO are also presented Table 2. In the free-living calibration, the wrist position displayed the lowest correlation and agreement.

### 3.4. Workload

Associations between single-sensor acceleration and full-body acceleration are visualized in Figure 4 and R^2^ is presented in Table 2 (Workload). All sensor positions performed similarly in the laboratory setting, whereas the performance of the wrist sensor was substantially lower than the performance of the other sensors in the free-living setting. In addition, the direction of the association between the wrist sensor and full-body acceleration was substantially different between laboratory and free-living data.

## 4. Discussion

This is the first study to investigate the relationship between shoe-based accelerometry and EE. With regard to measuring EE, the performance of the shoe position is comparable to the most commonly used sensor positions, which are the hip and wrist positions. Furthermore, the free-living validation of the laboratory-generated calibration models provides deeper knowledge on how different sensor positions and calibration procedures affect the measurement of PA in a free-living setting.

In general, all sensor positions performed well in the laboratory setting, both with regard to EE and mechanical work, with R^2^ consistently above 0.9. Hence, no conclusions on the performance of the different sensor positions can be drawn by the laboratory calibration alone. In the validation part, on the other hand, the performance was more diverse, especially between the wrist sensor and the other positions investigated. In both the laboratory derived models and the models derived from free-living data, the correlation between measured and estimated EE was weaker for the wrist position compared to the shoe-, hip- and thigh sensors. Figure 2 and Figure 3 show how the estimated and measured EE are more randomly spread out with the wrist position compared to the other positions. The correlation between estimated and measured EE based on the free-living calibration was similar with the hip, thigh and shoe positions, whereas the wrist position performed worse when using free-living data for calibration. With regard to the agreement between estimated and measured EE on the other hand, the wrist position displayed the lowest error (RMSE) in the validation of the laboratory-calibrated model. Figure 3 suggests that this is mainly caused by the wrist position having the lowest mean bias of the positions investigated. The shoe-, hip- and thigh positions underestimated EE on most intensities except the lowest. However, with the calibration models derived from the free-living data, the wrist position displayed the largest error.

There is a clear difference between the wrist position and the other positions investigated with regard to the relationship with measured EE in the validation part. This difference is most likely caused by decoupling of the wrist sensor. Decoupling refers to when the movement of the wrist is not representative to the movement of the rest of the body [17]. The decoupling of the wrist sensor is apparent in Figure 4, where the direction of the association between wrist and full-body acceleration is different between laboratory and free-living data. The association between wrist and full-body acceleration is mainly driven by the difference in acceleration magnitude between the stationary and locomotive activities. However, since the participants were solving Sudoku during the stationary activities, some movement of the wrist was captured. This movement of the wrist during the stationary activities is apparent in Figure 4 and the direction of the association while only considering stationary activities is more in line with the free-living association. Varying degrees of decoupling of the wrist sensor explain the more randomly spread out samples of estimated and measured EE in Figure 2 and the lower correlations in general with wrist sensors.

The activity types of the present sample in the free-living setting have been reported previously [10]. Among the observed activities, 82% were considered either sitting or standing. Sitting and standing are typically considered sedentary activities and are associated with an EE of below 1.5 METs [2,18]. In the validation part of this study, the measured EE is considerably higher than 1.5 METs most of the time. However, this is in line with previous research, where sitting and standing activities in a workplace setting typically reach between 1.5 and 4.5 METs [18]. Similarly, when comparing previously published accelerometer cut-points for EE, cut-points retrieved from calibrations based on locomotive activities are higher than cut-points retrieved from household activities [19,20]. Previously published cut-points using the same data processing methods as in the current study have all been using locomotive activities for calibration and the cut-points are similar to the cut-points from the laboratory calibration in the current study [7,14]. This explains the lower acceleration cut-offs for the hip, thigh and shoe sensors from free-living data compared to laboratory data in Table 3.

In the free-living validation of the current study, EE is in the range between the stationary activities and walking (Figure 1). The movement captured by the hip, thigh and shoe accelerometers is lower in the free-living stationary activities compared to the in-laboratory locomotive activities. With the wrist sensor, the captured acceleration is more similar between laboratory and free living. This explains why hip, thigh and shoe sensors tend to underestimate EE in free living (Figure 2). Furthermore, this also explains why the wrist sensor displayed the lowest error with the laboratory calibration and why the performance of the other positions was better when calibrating the model on the free-living data.

We have only found one previous study comparing the performance of different sensor positions in free living [9]. In this study, the wrist position performed better than the thigh position when considering doubly-labeled water as the gold standard for EE. Doubly-labeled water allows longer measurements of EE, but lacks the resolution that indirect calorimetry provides [21]. However, the calibration procedure was not comparable to the present study. The calibration of the wrist output was performed using free-living data, with a multisensory solution combining a trunk positioned accelerometer and heart rate monitoring as reference [22]. The thigh sensor, on the other hand, was calibrated based on a separate free-living sample where the wrist sensor was used as the only reference [9]. Although a free-living calibration of the wrist sensor might be positive, indirect calorimetry used in the present study is a better option for reference. Furthermore, the calibration of the thigh sensor using the wrist sensor output as reference is problematic, especially when the aim was to compare the two positions. Other validation studies performed separately using different sensor positions suggest that the hip is superior to the wrist [23,24]. However, the performance varies with different processing methods of accelerometer data, which limits comparability between studies [25].

The use of hip sensors was originally motivated by the position being close to the body’s center of mass [26]. A positioning close to the center of mass was proposed since this should represent the whole body movement as accurately as possible when just using one sensor. The present study supports this by showing the strongest relationship between single sensor and full body acceleration in free living with the hip position. Interestingly the thigh and shoe positions are more representative of the full body movement compared to the wrist position, despite being more distal to the center of mass. This could be explained by the shoe and thigh sensors capturing acceleration related to ground reaction forces reasonably well [27]. The intensity specific cut-points presented in Table 3 are decreasing from shoe to thigh and hip the further away from the ground the movement is measured. This is due to the dampening effect of the joints that the ground reaction force travels through from the ground and upwards [13].

A possible solution for improving the prediction of EE from accelerometer data is the use of machine learning by artificial neural networks (ANN) for calibration. ANN enables utilization of more information from the accelerometer signal than only the magnitude of the movement [28]. The performance of wrist sensors have been improved substantially by applying ANN calibration [29]. However, ANN calibration does not seem to improve the performance of hip and thigh sensors. Since the performance of the shoe sensor was similar to the hip and thigh sensor in the present study, it is unlikely that this performance would be substantially improved by ANN calibration. Nevertheless, most large-scale studies using wrist sensors analyze PA intensity by simple cut-points derived in a similar way as in the current study [30,31].

Previously, shoe sensors have mainly been used in PA measurement for activity classification. These classification algorithms have been developed using machine learning techniques such as decision trees and random forest classifiers [10,11,32]. In general, the accuracy of activity classification methods is well above 90% when cross-validated on unseen data from the same dataset that was used for calibration. Similar to the results in the current study, different sensor positions all perform similarly well in these settings [32]. However, the accuracy drops substantially when the models are validated in a separate dataset of structured activities [33], and even more so when validated in a free-living setting [10]. In addition, thigh sensors are more commonly used for activity classification [8]. Thigh sensors are especially useful for distinguishing sitting from standing, since the angle of the thigh differs between the activities. Activity classification could be used in combination with EE estimation in order to get at deeper understanding of the PA intensity of sitting and standing, which could be of particular importance in workplace settings [34].

A limitation apparent in the current study is that the activities included in the calibration protocol were different from the activities performed by participants in the free-living validation. In addition to sitting and standing work, participants were instructed to both walk slow and walk fast. However, the slow walking performed was in most cases more intense than the intensity of any of the free-living activities. Therefore, activities that are more diverse should be used in the calibration protocol. Nevertheless, the recommended and most commonly used calibration cut-points mainly use locomotive activities for calibration [6]. The calibration models developed from the validation data overcome this limitation to a large degree, but come with a couple of other limitations. The validation data based on participant mean provide relatively few data points since there were only 15 participants. The minute-by-minute measurements, on the other hand, provide much more data points, but EE does not respond instantaneously to mechanic work [12]. In the present study, this has been addressed by comparing the minute-by-minute average EE to the average acceleration of the current and previous minute. Figure 2 and Figure 3 suggest that the minute-by-minute measurements are similar to the participant mean and that this way of analyzing the data is reasonable. Moreover, it is problematic to validate the calibrations derived from the free-living dataset in the same dataset, although LOO was applied. LOO handles the issue with individual variation, but not the issue with the specific activities at the current workplace. This limits generalizability to other free-living settings. A further limitation of the current study is that the two workplace setting groups were on average older and had a higher BMI than the laboratory group. There was also a larger proportion of men in the industrial production group than in the laboratory group. These differences could have contributed to decreasing the performance of the laboratory-derived calibrations in the free-living setting. There is previous research suggesting that the relationship between accelerometer output and energy expenditure could differ between men and women [35].

The research about the relationship between occupational PA and health is inconclusive. There are conflicting results, where some studies suggest occupational PA is favorable for workers health and some studies suggest it might be detrimental [3]. One proposed explanation is that fitness plays a key role in the relationship by high fitness reducing the risks associated with high workload [36]. Nevertheless, more high quality measurements of occupational PA are required to investigate this relationship further. However, with occupational PA measurements becoming more widespread, ethical problems with surveillance of workers should be considered [37].

## 5. Conclusions

The shoe-positioned accelerometer performed similarly to the most commonly used sensor positions when measuring PA in a workplace setting. The performance of the shoe sensor was more similar to the hip and thigh positions, whereas the wrist position performed differently with regard to measuring laboratory and free-living activities. Using calibrations derived from laboratory data for estimating EE in free living, the hip, thigh and shoe positions displayed stronger correlations but weaker agreement between estimated and measured EE compared to the wrist position. When free-living data was used for calibration, the agreement from the hip, thigh and shoe sensors improved and outperformed the wrist sensor. This study highlights limitations in using the relationship between accelerometer output and EE from a laboratory setting to estimate EE in a free-living setting. Free-living calibrations appear superior to laboratory calibrations. Furthermore, the results suggest that shoe-based accelerometers could be suitable for PA measurement in workplace settings.

## Figures and Tables

**Figure 1 sensors-21-02333-f001:**
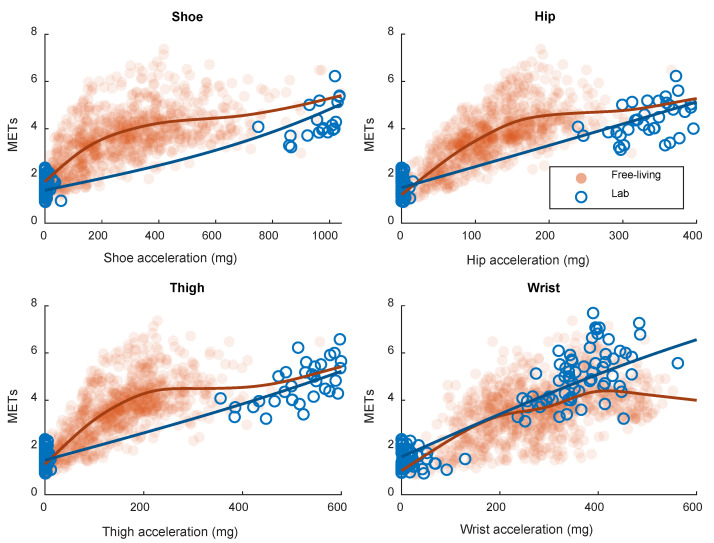
Calibrations between accelerometer output and energy expenditure (METs) in laboratory and free living based on smoothing splines.

**Figure 2 sensors-21-02333-f002:**
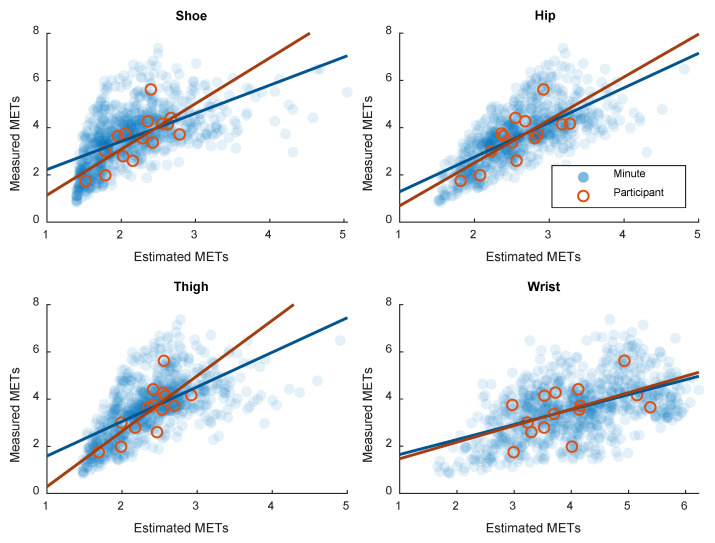
Correlation plots between estimated and measured energy expenditure (METs) based on minute-by-minute measurement and participant mean (free-living part).

**Figure 3 sensors-21-02333-f003:**
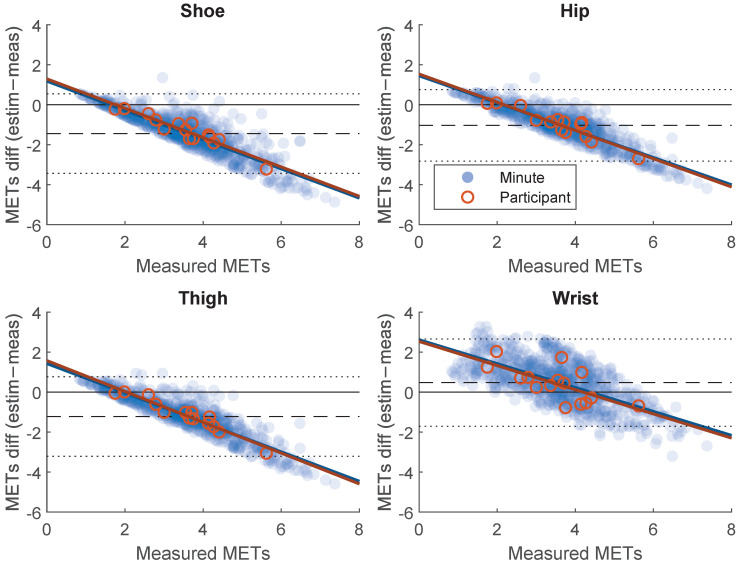
Bland–Altman plots of the agreement between estimated and measured energy expenditure (METs) based on minute-by-minute measurement and participant mean (free-living part). Dashed line represents mean difference and dotted lines represent limits of agreement (±2 SD).

**Figure 4 sensors-21-02333-f004:**
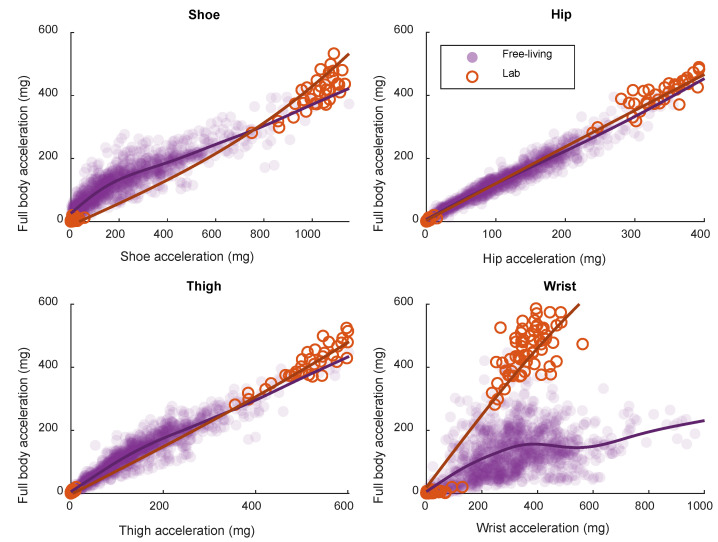
Association between single-accelerometer output and full-body acceleration in laboratory and free-living, based on smoothing splines.

**Table 1 sensors-21-02333-t001:** Participants’ characteristics.

	*N* (Female %)	Age (SD)	BMI (SD)
Laboratory	34 (47%)	25.4 (6.1)	23.1 (2.3)
Logistics warehouse	15 (20%)	39.6 (11.9)	27.0 (3.9)
Industrial production	14 (0%)	37.6 (11.5)	26.5 (3.1)

**Table 2 sensors-21-02333-t002:** Performance of calibration models in laboratory and free living (95% CI).

		Shoe	Hip	Thigh	Wrist
Calibration					
Laboratory	R^2^	0.91 (0.93–0.95)	0.93 (0.92–0.96)	0.90 (0.91–0.95)	0.91 (0.85–0.92)
Free living	R^2^	0.44 (0.33–0.42)	0.54 (0.46–0.56)	0.50 (0.39–0.49)	0.30 (0.23–0.32)
Validation of laboratory calibration					
Subject mean	r	0.72 (0.33–0.90)	0.73 (0.33–0.91)	0.73 (0.33–0.91)	0.53 (0.02–0.82)
	RMSE (METs)	1.49 (1.16–2.07)	1.23 (0.90–1.73)	1.40 (1.04–1.99)	0.94 (0.67–1.33)
Minute-by-minute	r	0.57 (0.52–0.61)	0.68 (0.64–0.72)	0.62 (0.58–0.66)	0.50 (0.45–0.55)
	RMSE (METs)	1.75 (1.69–1.83)	1.37 (1.31–1.43)	1.58 (1.51–1.65)	1.19 (1.14–1.24)
Validation of free-living calibration (LOO)					
Subject mean	r	0.64 (0.19–0.87)	0.72 (0.24–0.91)	0.67 (0.15–0.90)	0.34 (−0.21–0.73)
	RMSE (METs)	0.74 (0.52–1.15)	0.74 (0.52–1.09)	0.71 (0.43–1.23)	0.92 (0.66–1.24)
Minute-by-minute	r	0.58 (0.54–0.63)	0.66 (0.62–0.70)	0.64 (0.60–0.68)	0.39 (0.33–0.45)
	RMSE (METs)	0.97 (0.93–1.03)	0.88 (0.83–0.92)	0.95 (0.90–1.00)	1.12 (1.06–1.17)
Workload					
Laboratory	R^2^	0.97 (0.98–0.99)	0.99 (0.99–0.99)	0.98 (0.98–0.99)	0.97 (0.95–0.97)
Free living	R^2^	0.83 (0.79–0.85)	0.95 (0.94–0.95)	0.90 (0.88–0.91)	0.24 (0.25–0.34)

R^2^ explained variation, r Pearson correlation coefficient, RMSE root mean squared error, METs metabolic equivalents, LOO leave one out cross validation.

**Table 3 sensors-21-02333-t003:** Accelerometer cut-points for energy expenditure.

Energy Expenditure	(METs)	Shoe (mg)	Hip (mg)	Thigh (mg)	Wrist (mg)
Laboratory calibration	1.5	43	1	7	1
	3	575	168	265	154
	6	1201	494	701	519
	9	1623	816	1039	1038
Free-living calibration	1.5	1	12	10	36
	3	122	75	89	158

## Data Availability

The data presented in this study are available in Appendix A.

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
