# Peer review of "Measurement of Physical Activity by Shoe-Based Accelerometers—Calibration and Free-Living Validation"

_sensors, 2021, doi:10.3390/s21072333_

Round 1

Reviewer 1 Report

This study aims to develop calibration models to estimate energy expenditure from shoe-based accelerometers, validate the performance in a workplace setting, and compare it to the most commonly used accelerometer positions. The study needs minor corrections (see my comments below).

1. Introduction:
The introduction should include a description of the impact of this study compared to the existing literature. 

2. Methods:
- Table 1 should be moved to the Results section.
- Please clarify subjects inclusion/exclusion criteria.

3. Results:
- Do subject characteristics could have a significant difference (e.g. age, gender)?

4. Conclusions:
Make a conjecture of what this study suggests in a broader scope in the Conclusion section.

Reviewer 2 Report

The manuscript “Measurement of Physical Activity by Shoe-based Accelerometers –Calibration and Free-living Validation” introduced a method to measure physical activity by shoe-based accelerometers. The accelerometer was set to be different positions including shoe, hip, thigh, and wrist to calibrate the energy expenditure (EE). The results indicated that the performance of the shoe position is comparable to other commonly used positions. Besides, using calibrations derived from lab-data for estimating EE in free-living, the hip, thigh, and shoe positions displayed stronger correlations but a weaker agreement between estimated and measured EE compared to the wrist position. This work highlighted the relationship between accelerometer output and EE from a lab setting. The topic is interesting, and the manuscript is well written and organized, which could be published as a current version.

Author Response

Thank you for the comments. 

Reviewer 3 Report

Abstract

Delete 'work place', add 'workplace (2x).

Delete 'lab', add 'laboratory' (3x). Also, all future occurrences throughout the article.

Line 13, add 'the' before shoe.

Is the term 'free-living' the most appropriate descriptor (in title and abstract)? Would 'non-laboratory or 'workplace' describe this better? Also, all future occurrences throughout the article.

Introduction

Line 31, delete 'occupational', add 'workplace'.

Line 48, delete 'work place', add 'workplace.

PA measurements have been conducted on animals such as greyhounds where the accelerometer was placed between the shoulder-blades.  https://www.nature.com/articles/s41598-020-63678-1.

Materials and Methods

Tabel 1 caption, delete 'characteristics', add 'gender, age and BMI'.

Table 1, delete 'N (% female)', add 'N (female %)'.

Line 83, add 'a' before 'self-selected'.

Line 94 and 95, delete '+/-', add '$\pm$'

Discussion

Line 265, add 'We also found two studies that used accelerometers to measure canines [https://www.nature.com/articles/s41598-020-63678-1].

It is also suggested that the discussion section be broken down into sub-sections such as 'validation etc.
